# Ear wound healing in MRL/MpJ mice is associated with gut microbiome composition and is transferable to non-healer mice via microbiome transplantation

Cassandra Velasco[1], Christopher Dunn[1], Cassandra Sturdy[2], Vladislav Izda[2], Jake Martin[2], Alexander Rivas[3], Jeffrey McNaughton[1], Matlock A. Jeffries[1,2]*

1 University of Oklahoma Health Sciences Center, Department of Internal Medicine, Division of Rheumatology, Immunology, and Allergy, Oklahoma City, Oklahoma, United States of America, 2 Oklahoma Medical Research Foundation, Arthritis & Clinical Immunology Program, Oklahoma City, Oklahoma, United States of America, 3 University of Arkansas for Medical Sciences, Little Rock, Arkansas, United States of America

* matlock-jeffries@omrf.org

**Data Availability Statement:** All 16S sequencing files are available from the NCBI Sequence Read Archive (SRA), accession number: PRJNA738540.

## Abstract

### Objective

Adult elastic cartilage has limited repair capacity. MRL/MpJ (MRL) mice, by contrast, are capable of spontaneously healing ear punctures. This study was undertaken to characterize microbiome differences between healer and non-healer mice and to evaluate whether this healing phenotype can be transferred via gut microbiome transplantation.

### Methods

We orally transplanted C57BL/6J (B6) mice with MRL/MpJ cecal contents at weaning and as adults (n = 57) and measured ear hole closure 4 weeks after a 2.0mm punch and compared to vehicle-transplanted MRL and B6 (n = 25) and B6-transplanted MRL (n = 20) mice. Sex effects, timing of transplant relative to earpunch, and transgenerational heritability were evaluated. In a subset (n = 58), cecal microbiomes were profiled by 16S sequencing and compared to ear hole closure. Microbial metagenomes were imputed using PICRUSt.

### Results

Transplantation of B6 mice with MRL microbiota, either in weanlings or adults, improved ear hole closure. B6-vehicle mice healed ear hole punches poorly (0.25±0.03mm, mm ear hole healing 4 weeks after a 2mm ear hole punch [2.0mm—final ear hole size], mean±SEM), whereas MRL-vehicle mice healed well (1.4±0.1mm). MRL-transplanted B6 mice healed roughly three times as well as B6-vehicle mice, and half as well as MRL-vehicle mice (0.74±0.05mm, $P$ = 6.9E-10 vs. B6-vehicle, $P$ = 5.2E-12 vs. MRL-vehicle). Transplantation of MRL mice with B6 cecal material did not reduce MRL healing (B6-transplanted MRL 1.3±0.1 vs. MRL-vehicle 1.4±0.1, p = 0.36). Transplantation prior to ear punch was associated with the greatest ear hole closure. Offspring of transplanted mice healed significantly better than

**Funding:** MJ: NIH grants K08AR070891, P20GM125528, R61AR078075, and R01AR076440, Congressionally Directed Medical Research Program grant PR191652, Presbyterian Health Foundation Physician Scientist Development Award (PDSA), Oklahoma Center for the Advancement of Science and Technology (HR16-066) Oklahoma Health Research Grant The funders had no role in study design, data collection and analysis, decision to publish, or preparation of the manuscript.

**Competing interests:** The authors have declared that no competing interests exist.

non-transplanted control mice (offspring:0.63±0.03mm, mean±SEM vs. B6-vehicle control:0.25±0.03mm, n = 39 offspring, P = 4.6E-11). Several microbiome clades were correlated with healing, including *Firmicutes* (R = 0.84, *P* = 8.0E-7), *Lactobacillales* (R = 0.65, *P* = 1.1E-3), and *Verrucomicrobia* (R = -0.80, *P* = 9.2E-6). Females of all groups tended to heal better than males (B6-vehicle *P* = 0.059, MRL-transplanted B6 *P* = 0.096, offspring of MRL-transplanted B6 *P* = 0.0038, B6-transplanted MRL *P* = 1.6E-6, MRL-vehicle *P* = 0.0031). Many clades characteristic of female mouse cecal microbiota vs. males were the same as clades characteristic of MRL and MRL-transplanted B6 mice vs. B6 controls, including including increases in *Clostridia* and reductions in *Verrucomicrobia* in female mice.

## Conclusion

In this study, we found an association between the microbiome and tissue regeneration in MRL mice and demonstrate that this trait can be transferred to non-healer mice via microbiome transplantation. We identified several microbiome clades associated with healing.

## Introduction

The Murphy Roths Large (MRL) mouse strain was originally selected for large body size in the 1960s; at the $F_{12}$ generation, these mice developed a spontaneous mutation in the immune regulator *Fas*. This MRL/*lpr* variant is commonly used as a lupus mouse model [1, 2], whereas the non-*Fas*-mutant MRL/MpJ strain are generally used as a control in autoimmune disease experiments. In 1998, MRL/MpJ mice were found to fully close 2.0mm ear hole puncture wounds after 4 weeks, contrary to wild-type mice that do not close ear hole wounds [3, 4]. Subsequent studies have shown a generalized healing phenotype in MRL mice, including neonatal digital tip regrowth [5, 6], peripheral nerve regeneration [7], and cardiac wound healing [8, 9].

Approximately 75% of the MRL genome is derived from the LG/J parent strain [10]. In experiments over the past decade, several backcrosses of LG/J and SM/J (a non-healer strain) have been generated to evaluate the heritability of ear hole closure and OA protection [11–14]. Those backcrosses most genetically similar to LG/J mice have the greatest regenerative ear hole wound healing capacity [15], as well as the highest articular cartilage regeneration ability [11, 16]. Genetic analyses of these strains have been conducted [10, 12, 17, 18], with several genes and gene pathways now known to be involved with wound healing [19], including DNA repair and Wnt signaling [12]. However, a 2015 report made an unexpected discovery: C57BL/6J mice did not improve either ear-wound or articular cartilage healing following an allogeneic bone marrow transplant from MRL/MpJ donors [20], and a recent study found the heritability of OA protection in LG/SM backcrosses to be only moderate (0.18 to 0.58) [21], suggesting that other environmental factors play a role in healing in MRL mice.

Of particular human relevance, MRL mice also regenerate knee cartilage following full-thickness cartilage injury, protecting them against the development of osteoarthritis (OA) [22]. Further studies have demonstrated a strong correlation between this articular cartilage healing phenotype and healing of ear puncture holes [11]. A number of MRL parent-strain backcross studies published over the past decade have shown a heritability and inflammatory component to both ear hole closure and OA protection in MRL mice [11–13]. OA is a chronic musculoskeletal disease characterized by pain and progressive loss of joint function leading to reduced mobility and is associated with a variety of comorbidities including heart disease,

stroke, metabolic syndrome, and anxiety/depression [23–25]. It is the most common musculo-skeletal disease worldwide and is the leading cause of chronic disability in the US, affecting roughly half of adults over 65 years of age [26]. There are no disease modifying drugs approved for OA; therefore, much attention has recently been focused on the development of novel regenerative strategies to treat this devastating disease.

In the present study, we set out to investigate whether the gut microbiome is associated with the cartilage healing phenotype observed in MRL mice, whether this trait is transferable to non-healer mice via alteration of the gut microbiome, and whether microbiome differences are associated with sex-associated discordant ear hole closure in MRL mice.

## Methods

### Ethics statement

The Oklahoma Medical Research Foundation institutional animal care and use committee (IACUC) approved this study (OMRF IACUC protocol #16–40, 19–43).

### Mouse husbandry

Young male and female (4 week-old) C57BL6/J or MRL/MpJ mice were purchased from Jackson Laboratory (Bar Harbor, ME, USA) and housed at the Oklahoma Medical Research Foundation (OMRF). Breeding pairs were created and pups were used for experiments. All animals were permitted ad libitum access to food and water (NIH31). The OMRF animal facility uses a 12-hour light-dark cycle. All animal husbandry procedures adhered to the NIH Guide for the Care and Use of Laboratory Animals. There were no adverse events (expected or unexpected) during the course of this experiment. The chow diet used for all mice in this study was the PicoLab Rodent Diet 20 (#5053 from LabDiet). This diet consists of 4.7% crude fiber, 5.0% fat (ether extract), 5.6% fat (acid hydrolysis), and 20.0% protein. Caloric content is 25.651% from protein, 13.205% from fat, and 62.144% from carbohydrate.

### Mouse cecal microbiota transplantation procedure

Cecal donor mice (B6 or MRL, 10–14 weeks of age) were sacrificed and immediately dissected under sterile conditions. The cecum was removed and cecal contents transferred to a sterile tube containing a 1:1 mixture of glycerol and phosphate-buffered saline (5mL), which was then filtered through a 100μM filter. The resulting mixture was aliquoted and frozen at -80C for subsequent transplantation. At 3–4 weeks of age, recipient mice were pretreated with omeprazole (50mg/kg body weight) as previously described [27, 28]. Omeprazole was then sequentially administered via oral gavage once daily for 3 days prior to microbiome transfer. On the day of cecal transplantation, 100uL of transplant material (per 3-4-week-old recipient mouse) or 300uL of transplant material (per 16-week-old adult mouse) was transplanted via oral gavage using flexible 30mm polypropylene tubes (Instech, Plymouth Meeting, PA, USA). Mice were then moved to clean cages and segregated by transplant group. For adult transplantation experiments, 12 week-old mice were given 50mg/kg omeprazole once daily for 3 days, then transplanted on day 4, as above.

### Mouse ear hole puncture, sacrifice procedures, ear hole measurement statistics

At 6 weeks of age (for young animals) or 18 weeks of age (for adult animals), mice were ear punched using a 2.0mm through-and-through ear punch. Ear punch size was measured to confirm 2.0mm initial size. Mice were sacrificed 4 weeks after ear punch (10 weeks of age for

young and 22 weeks of age for adult animals). Whole blood was collected via cardiac puncture, allowed to clot for at least 30 minutes, then centrifuged and serum removed for subsequent analysis. Final ear hole size was measured using digital calipers, investigators were blinded to mouse group during ear hole size measurements. Ear hole healing (mm) was then recorded as 2.0mm—final ear hole size; this healing value is reported throughout this article. Group differences in ear hole closure were calculated using a Student t-test, with $P<0.05$ considered statistically significant. Cecal material was collected immediately after sacrificing animals and flash frozen in liquid nitrogen. Cecal DNA was extracted using a QIAamp DNA microbiome kit (Qiagen).

## 16S ribosomal RNA (rRNA) gene sequencing

Microbial profiles were determined by sequencing a ~460bp region including the V3 and V4 variable region of bacterial 16S rRNA genes. The gene fragment was amplified from approximately 30ng of DNA in each sample (primers in S1 Table in S1 File) using a high-fidelity polymerase (NEB Q5, New England Biolabs) [29] and confirmed by 1% agarose gel electrophoresis. PCR master mixes were decontaminated with double-stranded DNAse treatment (PCR decontamination kit, Arcticzymes, Tromsø, Norway). Sterile water was processed using the same procedure as a negative control. Illumina Nextera XT indices were attached (Illumina), pooled in equimolar amounts, and sequenced on an Illumina miSeq sequencer using a 300bp paired-end sequencing protocol by the Clinical Genomics Center at OMRF.

## 16S rRNA OTU classification

Quality filtering, operational taxonomic unit (OTU) classification and microbial diversity analysis were performed using the Quantitative Insights into Microbial Ecology (QIIME) software package, version 1.9.1 [30]. Reads were clustered using the UCLUST algorithm [31] with a 97% pairwise identity threshold and taxonomy assigned using the GreenGenes 13_8 database [32].

## Diversity analyses

Alpha diversity was characterized using the observed OTUs method following rarefaction to the lowest number of OTUs present per group (123,543). Beta diversity was evaluated on a variance-adjusted, weighted unifrac model. Principal component analysis was performed and an Adonis (permuted analysis of variance, a multi-factor PERMANOVA) test with 999 permutations was used to calculate the statistical significance of group differences [33, 34].

## Group analyses

Group analyses were performed using the linear discriminant analysis effect size (LEfSe) pipeline [35]. LEfSe performs a non-parametric Kruskal-Wallis sum-rank test [36] to detect features with significant differential abundance between groups, $P \leq 0.01$ was considered significant. Next, it uses a linear discriminant analysis (LDA) [37] to estimate the effect size of each differentially abundant feature. An LDA threshold of $\geq 2$ was considered significant [38]. QIIME was used to calculate group Benjamini-Hochberg FDR-corrected q-values; $q \leq 0.01$ was chosen as the 'FDR-corrected' significance threshold. For Gram status comparisons, differences were evaluated by Student t-tests, $P \leq 0.05$ was considered statistically significant. Correlations were determined by comparing ear hole closure of individual animals with microbiome clades, $P \leq 0.05$ was considered statistically significant. No samples were excluded from analysis.

### Prediction of metagenome content and imputed bacterial functional classification

The Phylogenetic Investigation of Communities by Reconstruction of Unobserved States (PICRUSt) software package [39] was used to impute bacterial metagenomes from our 16S deep sequencing microbial DNA data, and functional annotation applied using the Kyoto Encyclopedia of Gene and Genomes (KEGG) catalog [40]. Statistical analysis was performed using the Statistical Analysis of Metagenomic Profiles (STAMP) package [41]. Statistical significance and effect sizes among the three groups (human OA-eroded, OA-intact, and control) were calculated using ANOVA. Statistical significance was defined as Benjamini-Hochberg FDR corrected $P \leq 0.01$.

### Sequencing data availability

Raw 16s microbiome sequencing data have been deposited to the NCBI Sequence Read Archive (SRA), submission identification number PRJNA738540.

## Results

### Improved ear hole closure in non-healer B6 mice transplanted with healer MRL cecal material

Our final analysis included 131 mice from 6 independent experiments. These included 25 B6 vehicle-transplanted control mice, 57 MRL-transplanted B6 mice, 29 MRL vehicle-transplanted control mice, and 20 B6-transplanted MRL mice. B6-vehicle mice healed ear hole punches poorly (0.25±0.03mm, ear hole healing 4 weeks after a 2mm ear hole punch [2.0mm —final ear hole size], mean±SEM), whereas MRL-vehicle mice healed well (1.4±0.1mm). MRL-transplanted B6 mice healed roughly three times as well as B6-vehicle mice, and half as well as MRL-vehicle mice (0.74±0.05mm, $P$ = 6.9E-10 vs. B6-vehicle, $P$ = 5.2E-12 vs. MRL-vehicle, Figs 1A and 2). Transplantation of MRL mice with B6 cecal material did not reduce MRL healing (B6-transplanted MRL 1.3±0.1 vs. MRL-vehicle 1.4±0.1, p = 0.36).

### Ear hole closure outcomes depend on the timing of microbiome transplantation

Next, we determined whether the timing of microbiome transplantation relative to ear punch would alter this healing phenotype. We performed ear punches of 6-week-old male B6 mice, similar to previous experiments, then performed an MRL cecal transplantation at 48 hours, 1 week, and 2 weeks after ear punch. There was a graded reduction in ear wound healing capacity in mice transplanted after ear punch (Fig 1B); the 48-hour post-ear punch group (n = 8) demonstrated superior healing compared to B6-vehicle mice (0.56±0.06mm vs. 0.25±0.03mm, P = 2.4E-5, n = 8), although there was a trend towards worse healing than mice transplanted at weaning (P = 0.11). In mice transplanted 1 week after ear punch, there was a trend towards improved healing (0.35±0.05mm, P = 0.11, n = 7). Healing was not improved compared to vehicle controls in mice transplanted 2 weeks after ear punch (0.26±0.05mm, P = 0.93, n = 6).

### Adult transplant recipients heal as well as weanling recipients

Although previous studies have indicated that the gut microbiome is most plastic in mice at weaning [42], the disease our laboratory studies (OA) is a disease of adulthood. Therefore, to broaden the applicability of our findings, we evaluated whether transplantation of mice as adults would still confer improvements in ear hole closure. To this end, we performed cecal

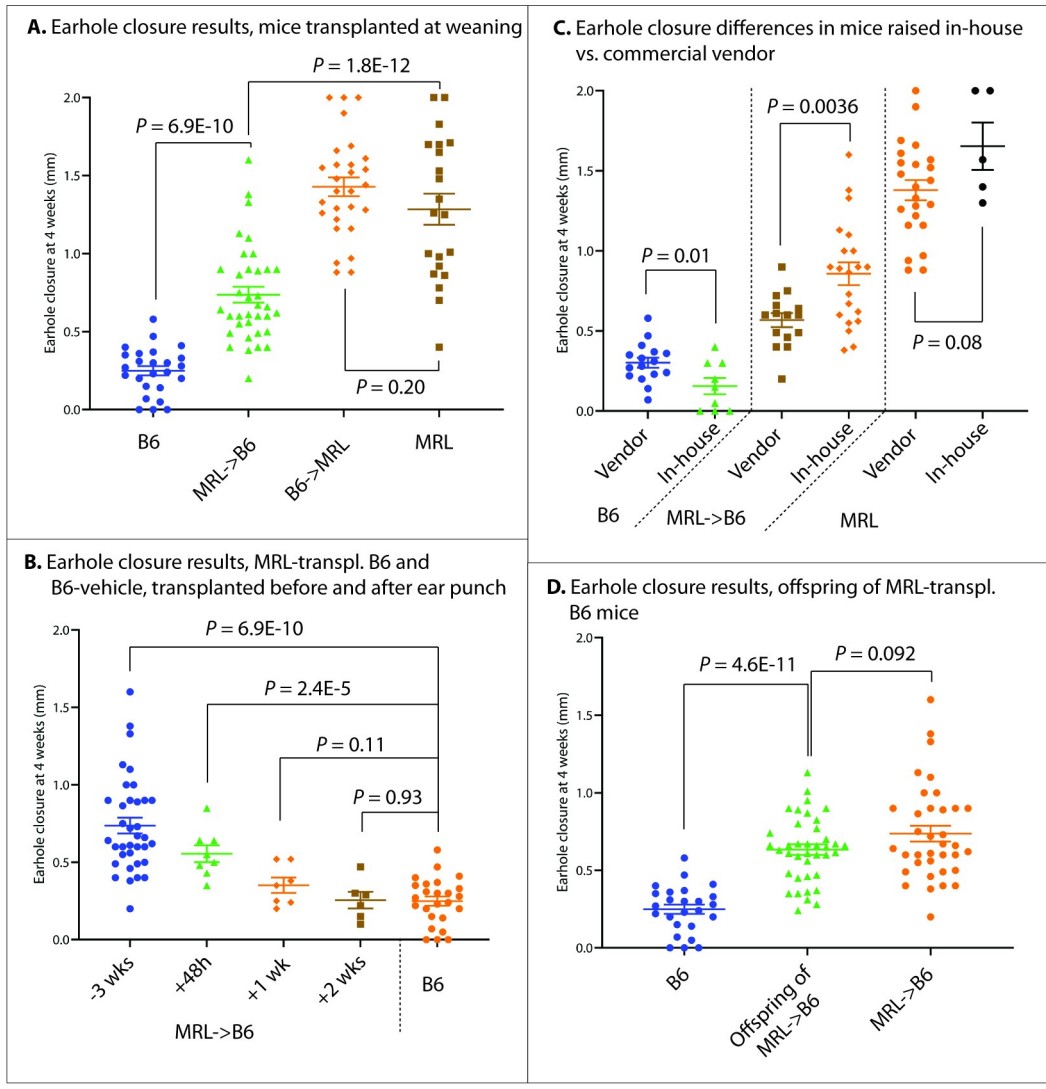

**Fig 1. Mouse earhole closure results 4 weeks after a 2.0mm ear punch under various microbiome transplantation conditions.**

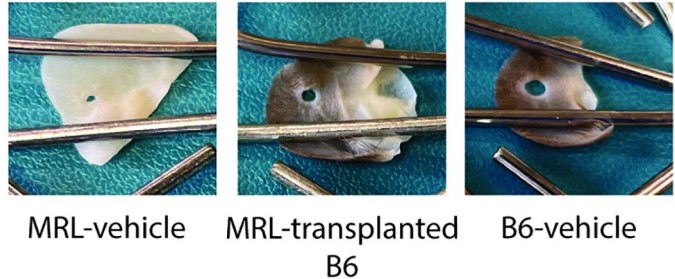

**Fig 2. Representative mouse ear images 4 weeks after a 2.0mm ear hole punch under various microbiome conditions.**

transplantation of B6 mice with MRL donor material (n = 7) or vehicle (n = 4), as previously described, in 12 week-old mice with ear hole punch 3 days later. No differences in ear hole closure were seen in mice transplanted at weaning compared to mice transplanted as adults (Fig 1C) (B6-vehicle weaning: 0.25±0.03mm ear hole closure vs. adult: 0.25±0.07, $P$ = 0.47; MRL-transplanted B6 weaning: 0.74±0.05 vs. 0.60±0.06, $P$ = 0.25).

## Distinct healing in mice raised in-house compared to mice sourced from a commercial vendor

Both the bacterial and viral components of the mouse gut microbiome are heavily influenced by diet and other environmental factors; in fact, previous studies have noted dramatic differences in the microbiome when comparing the same strain of mouse obtained from different commercial vendors [43, 44]. Roughly half of our mouse experiments were conducted on mice raised in-house (originally from Jackson laboratories, but bred for use at OMRF, including donors and recipient animals), with the other half being purchased commercially from Jackson laboratories and immediately used for transplantation experiments as both donors and recipients. We found variations ear hole closure with in-house reared mice demonstrating worse healing in the B6-vehicle group (commercial vendor: 0.31±0.03mm ear hole closure vs. in-house: 0.16±0.05, $P$ = 0.017) and improved healing in MRL-transplanted B6 (vendor: 0.57 ±0.04 vs. 0.86±0.07, $P$ = 0.0036), with a trend towards improved healing in MRL-vehicle mice (vendor: 1.4±0.06 vs. 1.7±0.1, $P$ = 0.082) (Fig 1D).

## Transgenerational heritability of microbiome-mediated improvements in ear hole closure

We then evaluated the heritability of improved ear hole closure following microbiome transplantation. We transplanted mice at weaning, as previously discussed, then created breeding pairs consisting of MRL-transplanted B6 male with MRL-transplanted B6 female, B6-vehicle male with MRL-transplanted B6 female, and MRL-transplanted B6 male with B6-vehicle female. The offspring of these breeding pairs were ear punched at 6 weeks of age and ear holes measured 4 weeks later to align with the timeline of our previous transplantation-at-weaning experiments. We found offspring of transplanted mice healed significantly better than non-transplanted control mice (offspring: 0.63±0.03mm, mean±SEM vs. B6-vehicle control: 0.25 ±0.03mm, n = 39 offspring, P = 4.6E-11); there was a trend toward slightly worsened healing compared to primary MRL-transplanted B6 mice (0.63±0.03mm vs. 0.74±0.05mm, P = 0.092) (Table 1, Fig 1E). There were no differences in healing among offspring where both sire and dam were transplanted compared to sire-only or dam-only-transplanted offspring (both transplanted: 0.65±0.02mm, n = 4, vs. one transplanted: 0.63±0.03, n = 35, P = 0.61).

## Durable gut microbiome alterations following microbiome transplantation, and microbiome clade associations with ear hole closure

Next, we profiled the cecal microbiomes of 43 animals, including 8 B6-vehicle (4 male and 4 female), 8 MRL-transplanted B6 (4 male and 4 female), 8 MRL-vehicle (4 male and 4 female), and 19 offspring of MRL-transplanted B6 mice (11 male and 8 female). All samples demonstrated high 16S sequencing quality, with raw read counts per sample ranging from 123,543 to 456,880. There were no statistically significant differences in raw read counts among any group. Importantly, we compared our B6 vehicle-transplanted mouse 16s findings from these experiments with wild-type B6 animals and did not identify any significant cecal microbiome shifts induced by the vehicle oral gavage or omeprazole pre-treatment (data not shown).

**Table 1.  Ear hole closure under various cecal transplantation conditions.**

| Primary microbiome transplantation | | | | | |
|---|---|---|---|---|---|
| **Mouse group** | **N** | **Ear hole closure (mm)** | **Ear hole closure P values vs. (sex-matched) B6-vehicle** | **Ear hole closure P values vs. (sex matched) MRL-vehicle** | **Ear hole closure P values, male vs. female** |
| **B6-vehicle** | **25** | **0.25±0.03** | | **6.9E-10** | **0.059** |
| Females | 12 | 0.31±0.04 | | 1.7E-10 | |
| Males | 13 | 0.19±0.04 | | 1.5E-12 | |
| **MRL-transplanted B6** | **36** | **0.74±0.05** | **6.9E-10** | **1.8E-12** | **0.096** |
| Females | 18 | 0.82±0.09 | 1.8E-7 | 2.5E-7 | |
| Males | 18 | 0.65±0.05 | 8.0E-5 | 1.8E-7 | |
| **B6-transplanted MRL** | **21** | **1.3±0.10** | | **0.20** | **1.6E-6** |
| Females | 8 | 1.77±0.06 | | 0.13 | |
| Males | 13 | 0.99±0.08 | | 0.025 | |
| **MRL-vehicle** | **28** | **1.4±0.07** | **6.9E-10** | | **0.0031** |
| Females | 15 | 1.5±0.10 | 4.0E-13 | | |
| Males | 13 | 1.3±0.08 | 2.4E-12 | | |
| Heritability: ear hole closure results of MRL-transplanted B6 mouse offspring | | | | | |
| **Mouse group** | **N** | **Ear hole closure (mm)** | **Ear hole closure P values vs.(sex-matched) B6-vehicle** | **Ear hole closure P values vs.(sex matched) primary MRL-transplanted B6** | **Ear hole closure P values, male vs. female** |
| **Offspring of MRL-transplanted B6** | **39** | **0.63±0.03** | **4.6E-11** | **0.092** | **0.0038** |
| Sire + Dam transplanted | 4 | 0.65±0.02 | 1.8E-5 | 0.71 | |
| Dam only transplanted | 23 | 0.65±0.04 | 1.1E-8 | 0.25 | |
| Sire only transplanted | 12 | 0.60±0.07 | 5.1E-6 | 0.16 | |
| Female offspring | 20 | 0.73±0.04 | 6.4E-7 | 0.35 | |
| Male offspring | 19 | 0.54±0.05 | 1.5E-6 | 0.10 | |

First, we analyzed alpha diversity, a measure of the overall number of bacterial clades present per group, using the observed_OTU method following rarefaction to the lowest read count (Fig 3A). MRL-vehicle samples demonstrated significantly higher alpha diversity than the other groups (MRL-vehicle 740±15 mean OTUs vs. B6-vehicle 600±30, P = 0.0014, MRL-vehicle vs. MRL-transplanted B6 640±32, P = 0.013, MRL-vehicle vs. offspring of MRL-transplanted B6 620±12, P = 5.4E-6). There were no statistically significant differences in alpha diversity among B6-vehicle, MRL-transplanted B6, and offspring of MRL-transplanted B6 mice. Beta diversity, a measure of diversity in the composition of groups, was then computed by the permanova method; we saw significant differences ($P<0.001$) among all groups (Fig 3B).

We found numerous differences in cecal microbial clades when comparing mouse groups (Fig 3C). In B6-vehicle vs. MRL-vehicle groups, 57 clades were significantly different (**S2 Table in** S1 File), in B6-vehicle vs. MRL-transplanted B6 groups, 17 clades were different (**S3 Table in** S1 File), in MRL-vehicle vs. B6-transplanted MRL groups, 239 clades were different (**S4 Table in** S1 File), and finally, comparing progeny of MRL-transplanted B6 mice to B6-vehicle mice, we found 55 differences (**S5 Table in** S1 File). A number of clade differences were shared among the various groups; for example, members of phylum *Verrucomicrobia*, particularly genus *Akkermansia*, were consistently characteristic of B6 mice or mice transplanted with B6 cecal content (LDA-ES 4.46, *P* = 0.0008 in MRL-vehicle vs. B6-vehicle comparisons; LDA-ES 4.23 with *P* = 0.0008 in MRL-transplanted B6 vs. B6-vehicle comparisons, LDA-ES 3.84, *P* = 0.04 in progeny vs. B6-vehicle, and LDA-ES 4.4 with *P* = 0.0008 in B6-transplanted MRL vs. MRL-vehicle comparisons). Conversely, phylum *Firmicutes* was consistently

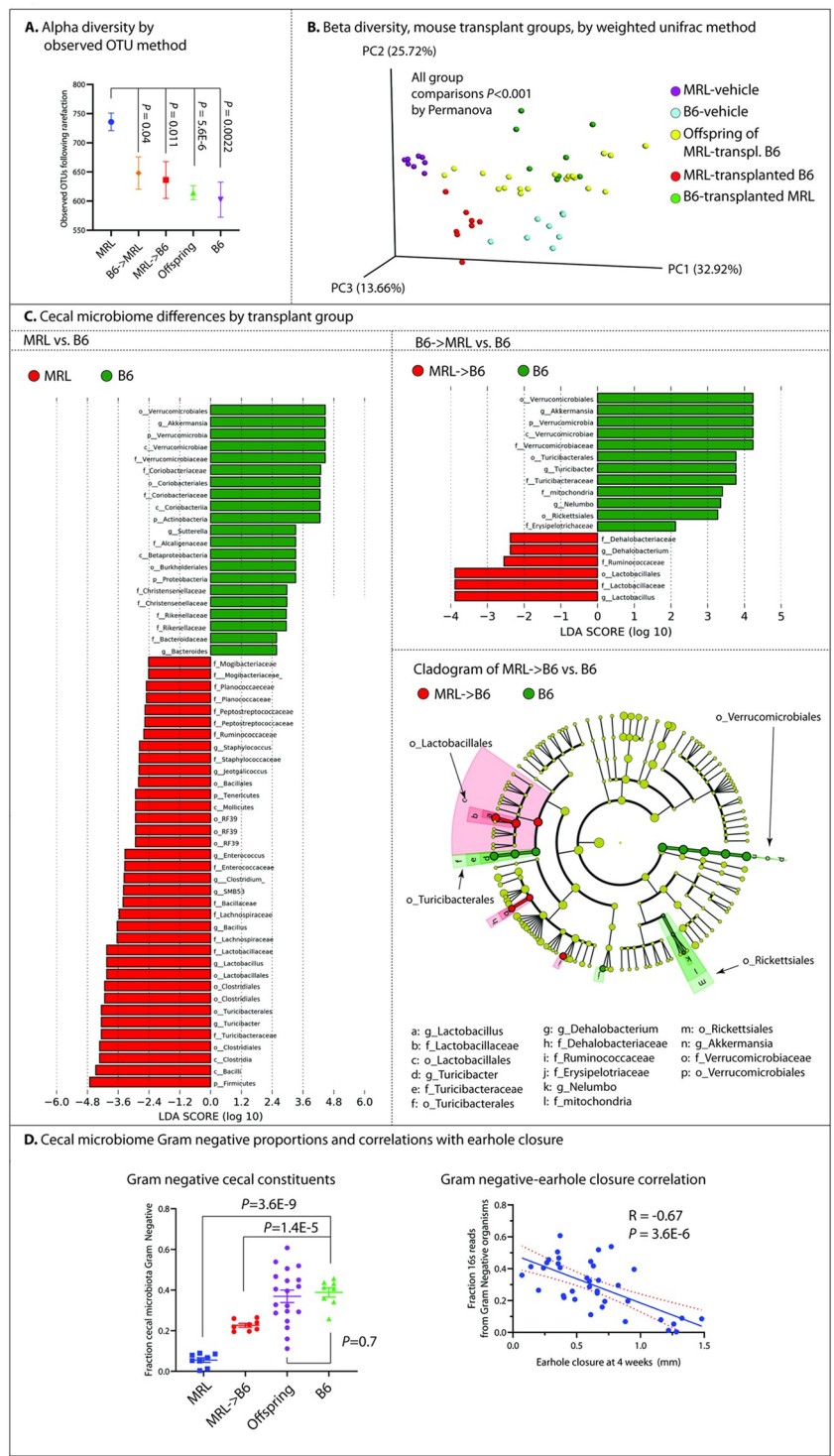

**Fig 3. 16S cecal microbiome analysis.**

associated with MRL or MRL-transplanted animals (LDA-ES 4.7, *P* = 7.8E-4 in MRL-vehicle vs. B6-vehicle; LDA-ES 4.2, *P* = 0.0030 in progeny vs. B6-vehicle; LDA-ES 4.6, *P* = 7.8E-4 in B6-transplanted MRL vs. MRL-vehicle). Within phylum *Firmicutes*, members of order *Clostridiales* were characteristic of MRL or MRL-transplanted animals (LDA-ES 4.32, *P* = 7.8E-4 in

MRL-vehicle vs. B6-vehicle; LDA-ES 3.76, *P* = 0.04; LDA-ES 4.42, *P* = 7.8E-4 in B6-transplanted MRL vs. MRL-vehicle), as were members of order *Lactobacillales* (LDA-ES 2.80, *P* = 7.8E-4 in MRL-vehicle vs. B6-vehicle; LDA-ES 3.89, *P* = 0.002 in MRL-transplanted B6 vs. B6-vehicle). Comparing B6-transplanted MRL to MRL-vehicle mice, we identified the transfer of B6-associated clades into recipient animals, including members of phylum *Verrucomicrobiae* (LDA-ES 4.40, *P* = 7.8E-4), whereas MRL-vehicle animals were again characterized by members of the phylum *Firmicutes* (LDA-ES 4.56, *P* = 7.8E-4), among others.

We next examined the Gram status of cecal contents in the present experiment. We found a decrease in Gram-negative constituent organisms in groups with improved ear healing (B6-vehicle fraction cecal reads Gram-negative 0.39±0.02 vs. MRL-transplanted B6 0.23±0.03 vs. MRL-vehicle 0.06±0.01, B6-vehicle vs. MRL-vehicle *P* = 3.6E-9, B6-vehicle vs. MRL-transplanted B6 *P* = 1.4E-5). However, offspring of MRL-transplanted B6 mice did not demonstrate a significant shift in microbiome Gram status (Offspring: 0.37±0.03, *P* = 0.70 vs. B6-vehicle). Nonetheless, individual Gram-negative bacterial fraction was negatively correlated with individual ear hole closure (Fig 3D, R = -0.67, *P* = 3.6E-6).

We then correlated microbial clades to individual ear hole closure among MRL-vehicle, B6-vehicle, and MRL-transplanted B6 mice. We found 48 clades highly correlated (Fig 4, **S6 Table** in S1 File), including positive correlations with phylum *Firmicutes* (R = 0.84, *P* = 8.0E-7), order *Clostridiales* (R = 0.76, *P* = 4.4E-5), and order *Lactobacillales* (R = 0.65, *P* = 1.1E-3). Negative correlations with ear hole closure were seen among phylum *Verrucomicrobia* (R = -0.80, *P* = 9.2E-6) and order *Burkholderiales* (R = -0.77, *P* = 2.3E-5). For many of these clades, significant group differences in abundance were seen among B6-vehicle vs. MRL-transplanted B6 (*Firmicutes*, *Lactobacillales*, *and Verrucomicrobia*) and all of these clades demonstrated significant group abundance differences comparing B6-vehicle to MRL-vehicle animals (Fig 4).

## Sex disparities in ear hole closure, and correlations with gut microbiome constituents

Female mice are less affected by surgically induced OA than are male mice [45]. As our study included both sexes, we evaluated sex-related differences in ear hole closure in control and transplanted mice, as well as any shared, sex-related microbiome differences that might underlie ear hole closure variations. Ear hole closure in MRL-transplanted B6 mice were not affected by the sex of the MRL donor mouse (*P* = 0.61), nor was closure in B6-transplanted MRL mice affected by the sex of the B6 donor mouse (*P* = 0.33). However, we did find variation in ear hole closure depending on the sex of the recipient mouse (Fig 5A). Female B6-vehicle demonstrated a trend towards an improved healing response (*P* = 0.059), with a similar trend found in female MRL-transplanted B6 mice (*P* = 0.096). Differences were seen in both female B6-transplanted MRL and female MRL-vehicle mice, where healing was significantly better than their male counterparts (*P* = 0.0015 and P = 4.2E-6 for B6-transplanted MRL and MRL-vehicle mice respectively). Furthermore, in heritability experiments, found a roughly one-third improvement in healing among female offspring compared to male offspring (*P* = 0.0038, Fig 5A).

For cecal 16S analysis, we pooled microbiome data from B6-vehicle, MRL-vehicle, and MRL-transplanted B6 animals and compared by sex. We first evaluated alpha diversity using the observed OTU method following rarefaction, as previously described, and did not find any sex-related differences (males: 646±13 vs. females: 632±21, *P* = 0.57). We found differences in beta diversity by sex both within mouse transplant groups (all *P*<0.001), and when all animals were considered together (*P* = 0.012) (Fig 5B). We then analyzed our 16S data by pooling all animals from the above groups and comparing male to female animals. We found 18 clades

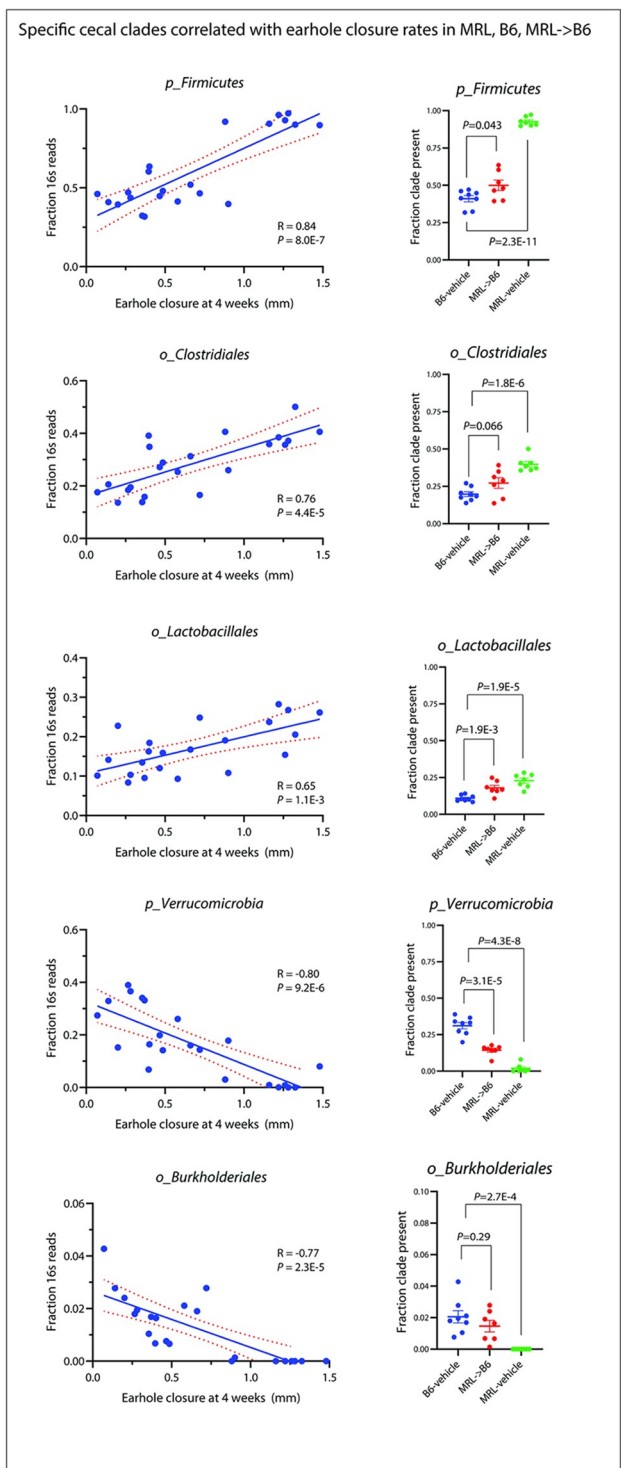

**Fig 4. Cecal microbiome correlations with earhole closure.**

which were different between the groups (Fig 5C); among these, many were shared with those clades identified as differentiating MRL and B6 animals, above. These included phylum *Verrucomicrobia* (increased in males, LDA-ES 3.88, *P* = 0.022) and order *Clostridiales* (increased in females, LDA-ES 3.99, *P* = 0.011) (**S7 Table in** S1 File).

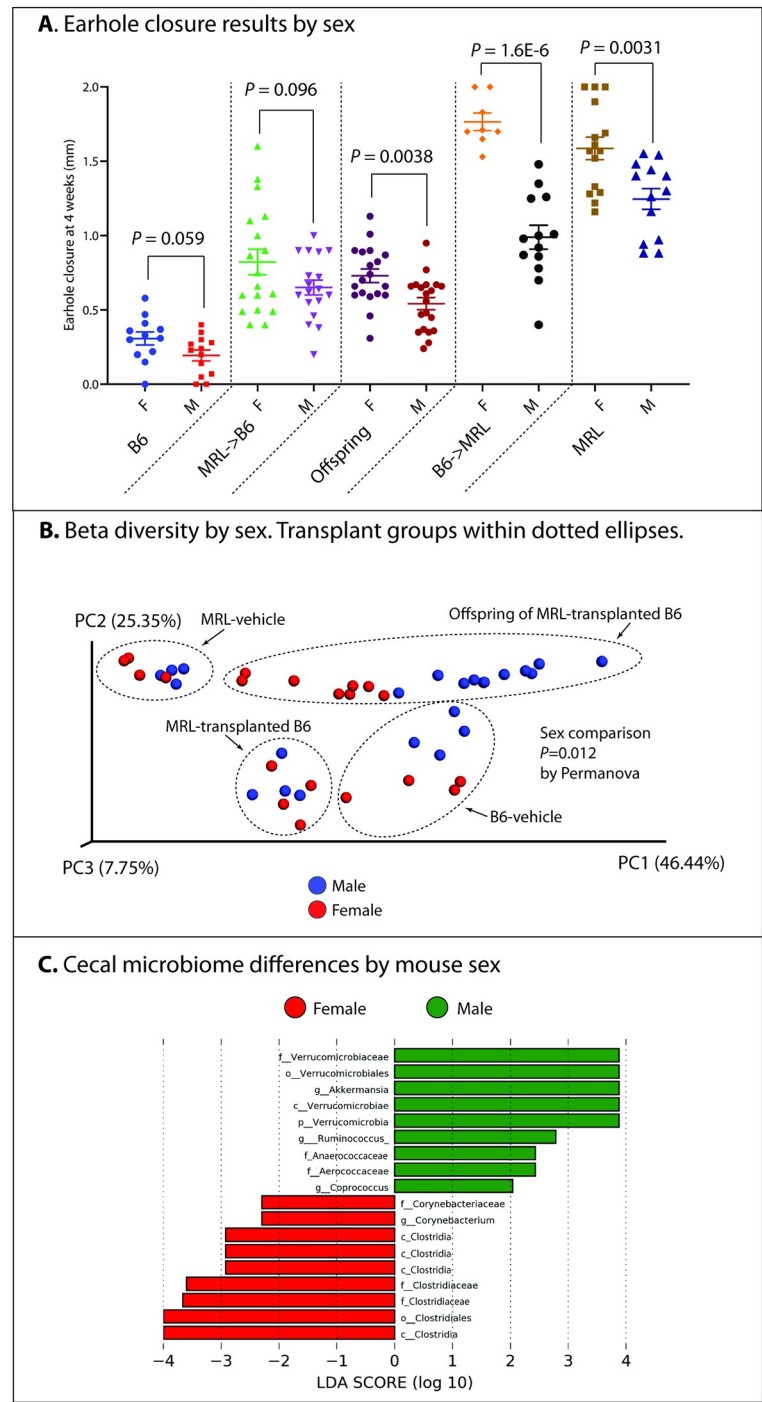

**Fig 5. Mouse earhole closure and cecal microbiome analysis by sex.**

## Imputed functional analysis of ear hole closure-associated gut microbiome changes

To offer insight into bacterial metabolomic and metagenomic changes associated with transplantation, we next imputed cecal microbiome metagenomic function using the Phylogenetic

**Table 2. Top imputed microbiome metagenomes correlated with ear hole closure among B6-vehicle, MRL-transplanted B6, and MRL-vehicle mouse groups.**

| KEGG pathway correlated with ear hole closure | Pearson R value | P value of correlation |
|---|---|---|
| Beta-Lactam resistance | -0.78 | 0.0027 |
| Penicillin and cephalosporin biosynthesis | -0.78 | 5.8E-4 |
| 1,1,1-Trichloro-2,2-bis(4-chlorophenyl)ethane (DDT) degradation | -0.77 | 0.0079 |
| Glycosphingolipid biosynthesis—ganglio series | -0.75 | 6.0E-4 |
| Lipopolysaccharide biosynthesis | -0.73 | 5.1E-4 |
| Meiosis—yeast | -0.73 | 4.3E-4 |
| Fatty acid elongation in mitochondria | -0.73 | 4.4E-4 |
| Proximal tubule bicarbonate reclamation | -0.73 | 3.2E-4 |
| Caffeine metabolism | -0.73 | 4.4E-4 |
| Fluorobenzoate degradation | -0.73 | 4.4E-4 |
| Circadian rhythm—plant | -0.73 | 4.4E-4 |
| Steroid biosynthesis | -0.73 | 4.4E-4 |
| Ubiquitin system | -0.72 | 4.8E-4 |
| Primary immunodeficiency | 0.56 | 0.0040 |
| Basal transcription factors | 0.61 | 0.0041 |
| Mineral absorption | 0.66 | 0.0030 |
| Flavone and flavonol biosynthesis | 0.72 | 0.0050 |
| D-Arginine and D-ornithine metabolism | 0.73 | 0.0065 |

Investigation of Communities by Reconstruction of Unobserved States (PICRUSt) analysis package (40). We compared MRL-transplanted B6 to B6-vehicle groups and identified 218 Kyoto Encyclopedia of Gene and Genomes (KEGG) pathways altered within gut microbiota (**S8 Table in** S1 File) at an FDR threshold of $P \le 0.01$. Of these, 48 were highly correlated with ear hole closure ($P \le 0.01$) (Table 2). The strongest positive correlations included D-arginine and D-ornithine metabolism ($R = 0.73$, $P = 0.0065$), flavone and flavonol biosynthesis ($R = 0.72$, $P = 0.0050$), and mineral absorption ($R = 0.66$, $P = 0.0030$), whereas negative correlations included beta-lactam resistance ($R = -0.78$, $P = 0.0027$), penicillin and cephalosporin biosynthesis ($R = -0.78$, $P = 5.8E-4$), glycosphingolipid biosynthesis ($R = -0.75$, $P = 6.0E-4$), and lipopolysaccharide biosynthesis ($R = -0.73$, $P = 5.1E-4$), among others.

## Discussion

In this study, we provide the first evidence that the spontaneous ear hole phenotype seen in MRL/MpJ mice is associated with specific gut microbiome constituents and is transferable to non-healer B6 mice via gut microbiome transplantation. Furthermore, we describe novel sex discrepancies in ear hole closure in B6 and MRL mice, where female animals demonstrate improved healing compared to males. We find that transplant-associated improvements in ear hole wound closure are time dependent, and that the best outcomes are achieved if gut microbiome transplantation occurs before ear wounding. Intriguingly, offspring of MRL-transplanted B6 mice also demonstrate improved healing compared to non-transplanted controls, indicating transgenerational heritability. In cecal microbiome 16S sequencing analysis, we identified a number of clades that are associated, both positively and negatively, with ear wound healing, including phylum *Firmicutes*, order *Clostridiales*, order *Lactobacillales*, and phylum *Verrucomicrobia*. We also found a negative association between Gram-negative organisms and ear wound healing despite an increase in serum LPS levels in both MRL and MRL-transplanted B6 animals.

These data offer an interesting addition to previously published data regarding genetic contributions to ear hole closure in MRL mice. Specifically, our data suggest that the unique genome of MRL mice may be influencing the microbiome, which, in turn, may be affecting ear hole closure outcomes. This interpretation is in line with previously published human data suggesting that human genetic makeup can influence the bacterial composition of the gut [46], and may affect the microbial composition of cutaneous wounds [47], which may have effects on wound healing. This interpretation is further supported by our lack of reduction in ear wound healing capability of MRL mice following transplantation of a B6 microbiome despite confirming a reduction in key healing-associated gut microbiome clades in B6-transplanted animals, suggesting genetic makeup may be able to 'override' the contributions of the microbiome in wound healing. Furthermore, although we noted transgenerational heritability of both the ear hole closure phenotype and cecal microbiota clades of MRL-transplanted B6 mice, we saw a trend towards reduction of ear hole closure in offspring compared to primary transplanted animals ($P$ = 0.09), as well as a reversion of Gram-negative gut constituents in offspring towards a 'baseline' B6 phenotype, suggesting that both the transplant-associated microbiome changes and cartilage healing phenotype may be lost over multiple generations. This raises the intriguing possibility that the unique genetic makeup of MRL mice is itself leading to alterations of the gut microbiome, perhaps through divergent immune responses within the gut, which subsequently contributes to the ear hole closure phenotype.

We also unexpectedly identified differences between male and female animals in both ear hole closure and gut microbiome constituent organisms. Previous studies have highlighted a reduction in OA histopathologic severity in female mice, in both young and old animals [45, 48]. Although not evaluated directly in the context of OA, sex-specific differences in the gut microbiome have been described in association with known OA risk factors, including high fat diet-induced obesity [49, 50] and advanced age [51]. In the present study, our 16S analysis identified 18 sex-associated microbiome clades; among these, 7 overlapped with our MRL vs. B6 comparison. In every case, healing-associated clades were enriched in females, whereas nonhealing-associated clades were enriched in males. If our ear hole closure phenotype indeed correlates with protection from OA in transplanted animals (to be evaluated in future studies), microbiome differences might offer a novel explanation for differences in sex-associated OA risk in mice. In this regard, our confirmation that the gut microbiome-mediated ear hole closure phenotype is stable even when transplanting adult animals bodes well for future studies of murine OA prevention.

Relatively few studies have rigorously evaluated the microbiome in the context of human cartilage regeneration through studies on OA. The largest to date, published in 2019 by Boer et al., found four bacterial clades associated with knee pain among 867 adults within the Dutch Rotterdam (RSIII) and LifeLines-DEEP OA cohorts [52]. These included class *Bacilli*, order *Lactobacillales*, family *Streptococcaceae*, and genus *Streptococcus*. In the present study, we found members of order *Lactobacillales* within the gut are strongly associated with ear wound healing in MRL-transplanted B6 mice, as well as one of the clades transgenerationally inherited to progeny of MRL-transplanted B6 mice. *Lactobacillus* has been shown to be depleted within the gut during mouse aging [53], a known risk factor for OA development. These findings align with our previous work which identified *Lactobacillales* DNA as characteristic of both disease-free human control and OA-protected MRL cartilage [54]. Similarly, we found family *Clostridiaceae* within cecal microbiota to be highly correlated with ear wound healing, and we previously identified this clade as associated with disease-free human cartilage. Members of the order *Burkholderiales* were strongly negatively correlated with ear hole closure; this order was associated with OA cartilage in our previous study. In 2019, Rios et al. published a report of decreased OA severity in a rat model following supplementation of a high fat/high sucrose

diet with prebiotic fiber; in their model, fiber treatment increased *Bifidobacterium* and *Roseburia* and decreased *Clostridium* and *Akkermansia* [55]. In our data, *Akkermansia* (within phylum *Verrucomicrobia*) was associated with poor healing, corroborating these findings.

The transgenerational heritability we identified in this study is interesting; previous reports have identified transgenerational heritability of microbiome-mediated inflammatory conditions [56], and a paper in 2020 demonstrated transgenerational heritability of weight gain, metabolic imbalance, and injury-induced OA for at least 2 generations following high-fat diet treatment in mice [57]. Unfortunately, this study did not evaluate the microbiome as a potential mediator of these effects. A number of studies have proposed an interaction between the gut microbiome and host epigenetic mechanisms in mediating transgenerational effects [58, 59]

Despite the associations we have described in this study, the key unanswered question is the mechanism driving cecal microbiome-mediated ear wound healing, which will no doubt require substantial future investigation to unravel. It is possible that alterations in the gut microbiome, through transplantation, may have induced systemic alterations in microbial metabolites and lead to a generalized pro-healing environment. Conversely, the metabolites of certain gut organisms, particularly short-chain fatty acids (SCFA), can promote intestinal repair and wound healing locally by inducing healing-associated gene expression in epithelial cells [60]. Another possible explanation would be the alteration of the cutaneous microbiome at the site of wounding, presumably via fecal-skin inoculation. Previous studies have indicated that altered skin flora plays a role in impaired cutaneous healing [61–63], principally related to biofilm formation at the wound site, with an associated increase in inflammatory cytokine production, limiting healing capacity [64].

Although we did not evaluate microbial metabolites directly, we did gain insight into metabolite alterations indirectly through our imputed functional metagenomic analysis. We found a variety of imputed KEGG pathways associated with ear hole closure, including a few with previous links to tissue regeneration/healing and OA development. These included an association with flavone biosynthesis, previously associated with promotion of wound healing in rats [65] and suggested as an anti-inflammatory agent for use in OA (by reduction in NF-KB and MAPK activation) [66]. We also saw an association with D-arginine and D-ornithine metabolism, both enhancers of wound healing in mice [67, 68]; intriguingly, there is some evidence that plasma arginine levels are reduced in knee OA patients [69]. Additional associations of interest include fatty acid elongation; serum fatty acid chain length is associated with symptomatic end-stage OA [70], and the bacterial secretion system, which enables the export of bacterial proteases to the extracellular environment in the context of chronic wound establishment [71].

Finally, our finding of differences in ear hole closure of animals raised in-house vs. those purchased from a commercial vendor and immediately used is in line with previous reports of cecal microbiome changes associated with various animal vendors [43, 44]. This observation also reinforces the potential for variation in microbiome-linked phenotypic changes like ear hole closure related to geographic location and diet; researchers should keep this in mind when designing future microbiome-related experiments.

Our study has several limitations. First, performed transplantation of animals with an already-existing microbiota rather than colonizing germ-free animals. Although using non-germ-free animals introduces the possibility of confounding results by variations in transplant engraftment, it also makes our study more compelling from a potential human therapeutic perspective. Another limitation is our lack of a definitive mechanistic explanation for the healing that we describe, this will no doubt require an extensive evaluation of several factors including local and systemic inflammatory responses, previously associated with healing in

MRL mice [72], and cell cycle/DNA damage response, also associated with healing in MRL and p21[-/-] mice [73]. Nevertheless, we felt that a timely publication of our observations was necessary to spur further research in this area. Finally, we limited our study to one spontaneous healer strain (MRL) and analyzed only young and young-adult mice; future analyses should include other strains and expand to evaluate the transplant-associated healing potential, if any, in older mice.

In summary, herein we offer the first description of the gut microbiome-associated nature of the ear hole cartilage healing phenotype previously described in MRL/MpJ mice, and demonstrate that this trait can be conferred to non-healer mice via a gut microbiome transplantation. We identified several clades within the gut microbiome that are strongly associated (both positively and negatively) with ear hole closure, and demonstrate that transgenerational inheritance of transplanted gut microbiome constituents and microbiome-mediate ear hole closure occurs. Furthermore, we found evidence for sex-specific effects of both ear hole closure and gut microbiome trends, with female animals healing better than males in a variety of contexts. Finally, we identified a number of differences in imputed metagenomes among healing-associated microbiome clades. Future work should focus on determining the mechanism underlying this microbiome-associated healing phenotype and expanding this analysis to include other animals identified as superhealers, including LG/J mice, *Acomys* mice, and perhaps invertebrate healers. The potential for a gut microbiome-mediated healing response should also be evaluated in the context of OA, which may offer a unique avenue for future therapeutic development.

## Supporting information

**S1 File.**
(XLSX)

## Author Contributions

**Conceptualization:** Cassandra Velasco, Christopher Dunn, Matlock A. Jeffries.

**Data curation:** Vladislav Izda, Matlock A. Jeffries.

**Formal analysis:** Cassandra Velasco, Christopher Dunn, Vladislav Izda, Jake Martin, Alexander Rivas, Jeffrey McNaughton, Matlock A. Jeffries.

**Funding acquisition:** Matlock A. Jeffries.

**Investigation:** Cassandra Velasco, Christopher Dunn, Cassandra Sturdy, Vladislav Izda, Jeffrey McNaughton, Matlock A. Jeffries.

**Methodology:** Cassandra Velasco, Christopher Dunn, Cassandra Sturdy, Jake Martin, Alexander Rivas, Matlock A. Jeffries.

**Project administration:** Cassandra Velasco, Matlock A. Jeffries.

**Resources:** Christopher Dunn, Matlock A. Jeffries.

**Software:** Christopher Dunn, Jake Martin, Matlock A. Jeffries.

**Supervision:** Christopher Dunn, Matlock A. Jeffries.

**Validation:** Cassandra Velasco, Christopher Dunn, Cassandra Sturdy, Vladislav Izda, Jake Martin, Alexander Rivas, Jeffrey McNaughton, Matlock A. Jeffries.

**Visualization:** Cassandra Velasco, Cassandra Sturdy, Vladislav Izda, Jake Martin, Alexander Rivas, Matlock A. Jeffries.

**Writing – original draft:** Matlock A. Jeffries.

**Writing – review & editing:** Cassandra Velasco, Christopher Dunn, Cassandra Sturdy, Vladislav Izda, Jake Martin, Alexander Rivas, Jeffrey McNaughton, Matlock A. Jeffries.

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
