## [Decision Letter · Decision Letter 0]

18 May 2021

PONE-D-21-05513

Ear wound healing in MRL/MpJ mice is associated with gut microbiome composition and is transferable to non-healer mice via microbiome transplantation

PLOS ONE

Dear Dr. Jeffries,

Thank you for submitting your manuscript to PLOS ONE. After careful consideration, we feel that it has merit but does not fully meet PLOS ONE’s publication criteria as it currently stands. Therefore, we invite you to submit a revised version of the manuscript that addresses the points raised during the review process.

Thank you for your patience with the handling of your manuscript.

Three reviewers found your study interesting and have brought up some concerns to address. For example, two of them have brought up the need to describe the diets of the controls, which can affect the baseline microbiome.

We look forward to receiving your revised manuscript.

Kind regards,

David M. Burmeister, PhD

Academic Editor

PLOS ONE

Additional Editor Comments (if provided):

Dr. Jeffries,

David

Journal Requirements:

2. As part of your revisions, please describe all measures undertaken to minimize potential pain and distress during your study. Additionally, please specify your monitoring parameters and humane endpoints. We thank you for your attention in this matter.

Reviewers' comments:

Reviewer's Responses to Questions

**Comments to the Author**

1. Is the manuscript technically sound, and do the data support the conclusions?

Reviewer #1: Yes

Reviewer #2: Partly

Reviewer #3: Yes

2. Has the statistical analysis been performed appropriately and rigorously? 

Reviewer #1: Yes

Reviewer #2: Yes

Reviewer #3: Yes

3. Have the authors made all data underlying the findings in their manuscript fully available?

Reviewer #1: Yes

Reviewer #2: No

Reviewer #3: No

4. Is the manuscript presented in an intelligible fashion and written in standard English?

Reviewer #1: Yes

Reviewer #2: Yes

Reviewer #3: Yes

5. Review Comments to the Author

Reviewer #1: Velasco et al. describes a series of experiments involving the use of two strains of laboratory mice, where the MRL/MpJ variant is apparently unique in the ability to heal cartilage in a short period of time (~4 weeks). The authors’ experiments investigate the role of the gut microbiome in the healing of ear cartilage punctures. The experiments involved a series of FMTs at different ages, inoculation times in relation to wounding, evaluation of heritability and with proper controls, including groups who received vehicle only. The analyses were well done and described. There were several notable findings that I read with great interest, including that FMT transplantation improved wound healing rate in B6 mice, benefits of FMT were less with delayed FMT, and those bacteria transferred by FMT could give advantages to the next generation. Particularly exciting to me, was the finding that gut microbiome composition correlated with external wound healing metrics. Perhaps this finding has been reported on elsewhere, but I am not aware of any prior research which documents this observation. Though I do give a few comments below on specific areas, there is clear merit to this study and the findings are communicated well.

Major comments:

There is just one analysis that I would like the authors to consider because I think it will be a brief and valuable analysis. The authors comment on a few taxa that were correlated positively or negatively with closure rate. How do these groups specifically vary in abundance in comparisons with and without FMT? Do you see that taxa that were correlated with improved closure also increase in FMT? I think that is a huge finding if you can further show that these taxa are correlated with healing AND those specific taxa were differentially abundant across FMT groups (preferably in a favorable direction).

Minor comments:

Lines 74-75: MRL/MpJ mice healing at 4 weeks, what was the previous expectation? I have little direct experience with model animals and some further context here could be useful.

Line 95: A period may be missing after first mention of OA.

Lines 113-117: Minor housekeeping issue, should the IRB protocol numbers be referenced here?

Lines 175-176: Usually when I read about rarefaction it seems to reference a number of reads rather than OTUs. Is this a typo? Please clarify.

Lines 280-282: Relating to previous comment, I see now here you are referring to the number of total OTU counts which is not technically wrong per se, but this is not typically referenced in this way and may confuse readers. To avoid confusion, I recommend swapping these specific mentions of ‘OTU’ with ‘read’. Reads are grouped into OTU clusters, and when I think of counting OTUs per sample, I’m thinking of unique OTUs per sample (of which it would be ridiculous to have 123,543 unique OTUs per sample).

I seem to recall an article that hit on the importance of mouse diet on studies like this, where labs trying to replicate others’ work sometimes had issues related to differences in mouse diet. I may have missed this, but can you describe the mouse diet in the relevant section of your methods? May have been related to fiber content but I can’t remember precisely right now. – Related, kudos to the analyses described in Figure 1D that I noticed after. Very nice inclusion.

Lines 400-416: I don’t know that I would cast out the role of genetics in these observations though clearly the microbiome seems to be doing something important here. There is a developing body of work that suggests that human genetics can influence the bacterial composition of the gut (https://www.nature.com/articles/s41564-020-0743-8) and even perhaps what microbes are able to colonize wounds (https://journals.plos.org/plospathogens/article?id=10.1371/journal.ppat.1008511). The wording may need to be toned down here, not because I disagree with you (I think it is probably a fair statement) but such statements would be better informed by some sort of genomic analysis and appropriate design that could rule out that the bacteria carried by the strain in question were not selected by some feature of mouse genomics.

Reviewer #2: The work presented in this paper seeks to understand how the gut microbiome contributes to ear hole healing in MRL mice. This is an interesting paper that raises the possibility that the healing trait in MRL mice is not fully genetically determined. This finding may also explain the failure of breeding studies to identify strong genetic determinants of tissue regeneration in the MRL mice.

The authors provide evidence that B6 mice transplanted with MRL cecal material partially healed ear hole wounds. The reverse experiment where B6 cecal material was transplanted into MRL mice did not change healing, despite the transfer of the B6 microbiome to recipient MRL mice. This finding weakens the argument that the microbiome contributes to healing phenotype. The possibility exists that the genetic control of hear hole closure ‘overrides’ microbiome control of regeneration in some way.

There are also issues with controls. A fuller description of controls, feeding regimes and baseline microbiome populations is needed. For example, it is not clear what the ‘vehicle-transplanted’ controls represent. Also, there was no attempt to establish the cecal population of each strain prior to any transplantation to establish a true microbiome baseline. The transplantation procedure may have introduced factors that influence subsequent flora populations.

For those wanting to repeat these studies, it will be important to understand in more detail the feeding regime and chow types used. This should be described in more detail.

It would have been useful to include a representative image of the extent of earhole closure for each experimental group.

Typo on line 429: “…..OA risk un mice.”

Reviewer #3: Abstract

The objective is a bit awkward. Authors start with cartilage but describe an ear wound phenotype and end up with regenerative therapy. MRL mice are also known to exhibit regenerative potential for articular cartilage and both ear wound and knee cartilage healing phenotypes are linearly correlated. The objective of the study here appears to be “to determine an association between gut microbiome and ear wound healing and to demonstrate whether this healing trait can be transferred to non-healer mouse by microbiome transplantation.” The point here is to clarify the objective and not to endorse the statement. Please consider as appropriate.

Line 50: “in mice” is not needed. Perhaps add a route of transplantation here i.e., oral.

C57BL/6J or B6, please use consistent terms or indicate acronyms at first appearance.

Results: Result section has no numeric data. Please include values to inform readers how much improvement is healing was observed (with mean +/- SD and P value). Add correlation coefficients for correlation studies and a P value. Some results are two descriptive. For example: “Females of all groups tended to heal better than males” AND “female microbiota resembled MRL mice”. It is not clear whether “better” means significant or not and if the effect size was low or high. Resembled in what sense. Was histology performed to measure tissue healing in situ? Please add numeric values to improve result section.

Conclusion: There is a disconnect in data presented and future directions. What is murine OA? Does it mean post-traumatic OA in mice, if so, then please change. What is the rationale of studying OA from ear wound healing phenotype? Is there a link between tissues regeneration and OA? Why not use these data to propose tissue regenerating studies in the ear and in hyaline cartilage?

Introduction

Some of words throughout the manuscript are unnecessarily connected: e.g., 2.0 mm should be 2 mm, earhole should be ear hole or ear-hole. same applies to earpunch, nonhealer…

Intro – first Para: authors have summarized various healing phenotypes in MRL but have not included ear wound and knee cartilage regeneration study by Sandell’s laboratory.

Line 79, please add a reference for 75% genome similarity between MRL and LG.

In experiments over the past decade, several backcrosses of LG/J and SM/J (a nonhealer

strain) have been generated to evaluate the heritability of earhole closure and OA protection. This is a strong statement but goes without any references. Please include all studies that utilized these strains in the last decade.

Line 91: MRL healing? Did author mean to say healing in MRL mice?

Line 95: period and reference lacking at the end of sentence.

Line 106: Authors only studied ear wound healing, so what does generally mean here?

Methods

Please confirm if IRB approval was needed for this kind of mouse studies. Please remove institutional review board approval if that does not apply since it relates to sampling human tissues.

It should be The Jackson Laboratory not laboratories.

Line 123: “some experiments” please specify.

In the abstract author pointed out 2.0 mm ear holes while 2 mm in methods. Please confirm if the hole diameter were confirmed to be 2.0 mm immediately after punching as not all whole created would be of same size.

Statistical analysis was not included separately but are included under other specific experiments. This is a bit unusual.

It is not clear in methods how ear hole data was reported. A larger hole would mean less healing but that does not appear to be the case. This needs to be clarified to avoid confusion.

Results

Representative images of ear wounds are not shown and should be presented.

244: disparate could be replaced by distinct.

“After a 2.0 mm ear punch” does not need to be repeated for all figures and tables.

In tables it is not clear why some values are bold and other non-bold.

Numeric data presentation does not follow a uniform pattern. P values and decimal points are not consistent throughout.

Discussion

Overall, the discussion is well-written. Please consider inclusion of a limitation that knee cartilage phenotype was not studied.

6. PLOS authors have the option to publish the peer review history of their article (what does this mean?). If published, this will include your full peer review and any attached files.

Reviewer #1: No

Reviewer #2: No

Reviewer #3: No

---

## [Author Response · Author response to Decision Letter 0]

24 May 2021

Please see the attached 'response to reviewers' document for a list of responses and corrections made in our revised manuscript.

Reviewer response letter

Dear PLOS ONE editors and reviewers:

Thank you very much for your insightful and thorough review of our manuscript, entitled “Ear wound healing in MRL/MpJ mice is associated with gut microbiome composition and is transferable to non-healer mice via microbiome transplantation”. We have addressed each of the issues raised by the 3 reviewers in our revised manuscript and have detailed discussions of each below. We believe that these revisions have significantly improved the quality of our manuscript and look forward to your evaluation.

Reviewer #1:

Velasco et al. describes a series of experiments involving the use of two strains of laboratory mice, where the MRL/MpJ variant is apparently unique in the ability to heal cartilage in a short period of time (~4 weeks). The authors’ experiments investigate the role of the gut microbiome in the healing of ear cartilage punctures. The experiments involved a series of FMTs at different ages, inoculation times in relation to wounding, evaluation of heritability and with proper controls, including groups who received vehicle only. The analyses were well done and described. There were several notable findings that I read with great interest, including that FMT transplantation improved wound healing rate in B6 mice, benefits of FMT were less with delayed FMT, and those bacteria transferred by FMT could give advantages to the next generation. Particularly exciting to me, was the finding that gut microbiome composition correlated with external wound healing metrics. Perhaps this finding has been reported on elsewhere, but I am not aware of any prior research which documents this observation. Though I do give a few comments below on specific areas, there is clear merit to this study and the findings are communicated well.

We thank the reviewer for this commentary regarding the novelty of our study.

Major comments:

There is just one analysis that I would like the authors to consider because I think it will be a brief and valuable analysis. The authors comment on a few taxa that were correlated positively or negatively with closure rate. How do these groups specifically vary in abundance in comparisons with and without FMT? Do you see that taxa that were correlated with improved closure also increase in FMT? I think that is a huge finding if you can further show that these taxa are correlated with healing AND those specific taxa were differentially abundant across FMT groups (preferably in a favorable direction).

We agree with this comment. There are several of these clades that were statistically significantly different in group comparisons, and others were not. We have changed our figures a bit, including a new Figure 3 to include additional highly correlated clades and included additional graphs for each of these clades demonstrating group differences with respective statistical values.

Minor comments:

Lines 74-75: MRL/MpJ mice healing at 4 weeks, what was the previous expectation? I have little direct experience with model animals and some further context here could be useful.

This finding was unexpected, as wild-type mice display quite poor wound healing (virtually no earhole closure). We have updated the text to reflect this.

Line 95: A period may be missing after first mention of OA.

Corrected, thanks.

Lines 113-117: Minor housekeeping issue, should the IRB protocol numbers be referenced here?

In line with comments from Reviewer 3, we have removed the IRB reference. We have also included the OMRF IACUC protocol numbers.

Lines 175-176: Usually when I read about rarefaction it seems to reference a number of reads rather than OTUs. Is this a typo? Please clarify.

We have updated the verbiage in this section.

Lines 280-282: Relating to previous comment, I see now here you are referring to the number of total OTU counts which is not technically wrong per se, but this is not typically referenced in this way and may confuse readers. To avoid confusion, I recommend swapping these specific mentions of ‘OTU’ with ‘read’. Reads are grouped into OTU clusters, and when I think of counting OTUs per sample, I’m thinking of unique OTUs per sample (of which it would be ridiculous to have 123,543 unique OTUs per sample).

Agreed, we have updated this verbiage to clarify.

I seem to recall an article that hit on the importance of mouse diet on studies like this, where labs trying to replicate others’ work sometimes had issues related to differences in mouse diet. I may have missed this, but can you describe the mouse diet in the relevant section of your methods? May have been related to fiber content but I can’t remember precisely right now. – Related, kudos to the analyses described in Figure 1D that I noticed after. Very nice inclusion.

The chow diet used was LabDiet’s PicoLab Rodent Diet 20 (stock #5053). We have included this information in the methods section.

Lines 400-416: I don’t know that I would cast out the role of genetics in these observations though clearly the microbiome seems to be doing something important here. There is a developing body of work that suggests that human genetics can influence the bacterial composition of the gut (https://www.nature.com/articles/s41564-020-0743-8) and even perhaps what microbes are able to colonize wounds (https://journals.plos.org/plospathogens/article?id=10.1371/journal.ppat.1008511). The wording may need to be toned down here, not because I disagree with you (I think it is probably a fair statement) but such statements would be better informed by some sort of genomic analysis and appropriate design that could rule out that the bacteria carried by the strain in question were not selected by some feature of mouse genomics.

We strongly agree and have revised this paragraph significantly to better reflect the “altered genetics -> microbiome changes -> earhole closure” relationship.

Reviewer #2: 

The work presented in this paper seeks to understand how the gut microbiome contributes to ear hole healing in MRL mice. This is an interesting paper that raises the possibility that the healing trait in MRL mice is not fully genetically determined. This finding may also explain the failure of breeding studies to identify strong genetic determinants of tissue regeneration in the MRL mice. The authors provide evidence that B6 mice transplanted with MRL cecal material partially healed ear hole wounds. The reverse experiment where B6 cecal material was transplanted into MRL mice did not change healing, despite the transfer of the B6 microbiome to recipient MRL mice. This finding weakens the argument that the microbiome contributes to healing phenotype. The possibility exists that the genetic control of hear hole closure ‘overrides’ microbiome control of regeneration in some way.

We agree and have changed our discussion paragraph, in line with comments from reviewers 1 and 3, to reflect this possibility. We disagree that our finding weakens the argument for a microbiome contribution to the healing phenotype, which we have shown clearly here. It does, however, weaken the argument for the microbiome being the sole determinant of earhole healing.

There are also issues with controls. A fuller description of controls, feeding regimes and baseline microbiome populations is needed. For example, it is not clear what the ‘vehicle-transplanted’ controls represent. 

Also, there was no attempt to establish the cecal population of each strain prior to any transplantation to establish a true microbiome baseline. The transplantation procedure may have introduced factors that influence subsequent flora populations.

We have actually performed these analyses (‘fully negative’ controls vs. vehicle-transplanted controls) but did not include them in the article. We have added a section within the results section to address this (lines 289-290).

For those wanting to repeat these studies, it will be important to understand in more detail the feeding regime and chow types used. This should be described in more detail.

We agree and have included this information in the methods section, in line with the comment from Reviewer 1.

It would have been useful to include a representative image of the extent of earhole closure for each experimental group.

 Added Supplementary Figure 1

Typo on line 429: “…..OA risk un mice.”

 Corrected, thanks.

Reviewer #3: 

Abstract

The objective is a bit awkward. Authors start with cartilage but describe an ear wound phenotype and end up with regenerative therapy. MRL mice are also known to exhibit regenerative potential for articular cartilage and both ear wound and knee cartilage healing phenotypes are linearly correlated. The objective of the study here appears to be “to determine an association between gut microbiome and ear wound healing and to demonstrate whether this healing trait can be transferred to non-healer mouse by microbiome transplantation.” The point here is to clarify the objective and not to endorse the statement. Please consider as appropriate.

Reworded.

Line 50: “in mice” is not needed. Perhaps add a route of transplantation here i.e., oral.

Corrected.

C57BL/6J or B6, please use consistent terms or indicate acronyms at first appearance.

Corrected.

Results: Result section has no numeric data. Please include values to inform readers how much improvement is healing was observed (with mean +/- SD and P value). Add correlation coefficients for correlation studies and a P value. Some results are two descriptive. For example: “Females of all groups tended to heal better than males” AND “female microbiota resembled MRL mice”. It is not clear whether “better” means significant or not and if the effect size was low or high. Resembled in what sense. Was histology performed to measure tissue healing in situ? Please add numeric values to improve result section.

Updated and expanded abstract to include numeric value.

Conclusion: There is a disconnect in data presented and future directions. What is murine OA? Does it mean post-traumatic OA in mice, if so, then please change. What is the rationale of studying OA from ear wound healing phenotype? Is there a link between tissues regeneration and OA? Why not use these data to propose tissue regenerating studies in the ear and in hyaline cartilage?

Corrected. As we describe within the manuscript text, there is a strong correlation between earhole closure rates and protection against post-traumatic osteoarthritis (OA) in mice, although this is probably too long to describe in detail in the abstract. We have updated our abstract to try to make this more clear.

Introduction

Some of words throughout the manuscript are unnecessarily connected: e.g., 2.0 mm should be 2 mm, earhole should be ear hole or ear-hole. same applies to earpunch, nonhealer…

 Updated

Intro – first Para: authors have summarized various healing phenotypes in MRL but have not included ear wound and knee cartilage regeneration study by Sandell’s laboratory.

Updated

Line 79, please add a reference for 75% genome similarity between MRL and LG. In experiments over the past decade, several backcrosses of LG/J and SM/J (a nonhealer strain) have been generated to evaluate the heritability of earhole closure and OA protection. This is a strong statement but goes without any references. Please include all studies that utilized these strains in the last decade.

Updated

Line 91: MRL healing? Did author mean to say healing in MRL mice?

Yes, corrected.

Line 95: period and reference lacking at the end of sentence.

Corrected.

Line 106: Authors only studied ear wound healing, so what does generally mean here?

Agree this is unclear, removed ‘generally’.

Methods

Please confirm if IRB approval was needed for this kind of mouse studies. Please remove institutional review board approval if that does not apply since it relates to sampling human tissues.

Removed the IRB mention.

It should be The Jackson Laboratory not laboratories.

Corrected.

Line 123: “some experiments” please specify.

Removed ‘some’. The purpose of this statement was to indicate the source of in-house bred animals (as opposed to commercial vendor-sourced animals).

In the abstract author pointed out 2.0 mm ear holes while 2 mm in methods. Please confirm if the hole diameter were confirmed to be 2.0 mm immediately after punching as not all whole created would be of same size.

A bit unclear what the reviewer means here, the hole size after punch (using a 2mm punch) was 2.0mm. We have updated the methods to further clarify.

Statistical analysis was not included separately but are included under other specific experiments. This is a bit unusual.

It is not clear in methods how ear hole data was reported. A larger hole would mean less healing but that does not appear to be the case. This needs to be clarified to avoid confusion.

Statistical methods used to define significance in earhole closure measurements were added to the Methods in the ‘Mouse earhole puncture, sacrifice procedures, earhole measurement statistics’ section.

Results

Representative images of ear wounds are not shown and should be presented.

 Added a supplementary image

244: disparate could be replaced by distinct.

Replaced.

“After a 2.0 mm ear punch” does not need to be repeated for all figures and tables.

Updated.

In tables it is not clear why some values are bold and other non-bold.

We bolded major group rows (i.e. B6-vehicle, MRL-transplanted B6, etc)., and left the by-sex breakdown rows below these as non-bolded.

Numeric data presentation does not follow a uniform pattern. P values and decimal points are not consistent throughout.

Numeric data have been updated to 2 significant digits throughout the manuscript.

Discussion

Overall, the discussion is well-written. Please consider inclusion of a limitation that knee cartilage phenotype was not studied.

We appreciate the reviewer’s comment; however, we do not feel a limitation of our current study was not evaluating a knee cartilage phenotype; the present study was focused solely on ear cartilage, future studies will certainly evaluate the knee healing phenotype.

---

## [Decision Letter · Decision Letter 1]

16 Jun 2021

PONE-D-21-05513R1

Ear wound healing in MRL/MpJ mice is associated with gut microbiome composition and is transferable to non-healer mice via microbiome transplantation

PLOS ONE

Dear Dr. Jeffries,

Thank you for submitting your manuscript to PLOS ONE. After careful consideration, we feel that it has merit but does not fully meet PLOS ONE’s publication criteria as it currently stands. Therefore, we invite you to submit a revised version of the manuscript that addresses the points raised during the review process.

Thank you for your proactive response to the reviewers, who have re-reviewed this manuscript and found only a couple of minor issues to fix.

Specifically, you will need to upload your raw sequencing data, which is often done with NCBI's SRA function- https://www.ncbi.nlm.nih.gov/sra, and a statement in the manuscript including the link/accession number will be needed. This is recommended for OA journals, but specifically required given your NIH funding.

Additionally, reviewer 3 had some important clarifications- please indicate how you came up with the wound healing numbers (right now it is assumed that remaining wound was subtracted form the original 2mm size). As reviewer 3 points out, technically this is not rate since the data are not longitudinal, which would be interesting data to report. At minimum, clarity in the methods would help.

We trust you will be able to address these concerns, and look forward to receiving your revised manuscript.

We look forward to receiving your revised manuscript.

Kind regards,

David M. Burmeister, PhD

Academic Editor

PLOS ONE

Journal Requirements:

Reviewers' comments:

Reviewer's Responses to Questions

**Comments to the Author**

1. If the authors have adequately addressed your comments raised in a previous round of review and you feel that this manuscript is now acceptable for publication, you may indicate that here to bypass the “Comments to the Author” section, enter your conflict of interest statement in the “Confidential to Editor” section, and submit your "Accept" recommendation.

Reviewer #1: All comments have been addressed

Reviewer #2: All comments have been addressed

Reviewer #3: (No Response)

2. Is the manuscript technically sound, and do the data support the conclusions?

Reviewer #1: Yes

Reviewer #2: Yes

Reviewer #3: Yes

3. Has the statistical analysis been performed appropriately and rigorously? 

Reviewer #1: Yes

Reviewer #2: Yes

Reviewer #3: Yes

4. Have the authors made all data underlying the findings in their manuscript fully available?

Reviewer #1: No

Reviewer #2: Yes

Reviewer #3: No

5. Is the manuscript presented in an intelligible fashion and written in standard English?

Reviewer #1: Yes

Reviewer #2: Yes

Reviewer #3: Yes

6. Review Comments to the Author

Reviewer #1: I am satisfied with changes made by the authors, however there is one item that needs to be addressed. In my previous review, I had mistakenly checked that data were available. The other two reviewers caught this correctly at that time. I do not see that data has been provided since then. For the Data Availability statement, the authors currently state that all data are available without restriction. The authors then point to the supplementary files which only contain some intermediate or raw result files of specific analyses. The standard in the field would be to publish the raw sequencing files and metadata on a database such as the Sequence Read Archive (SRA). At a minimum standard, authors need to provide raw sequencing files in fasta or fastq format as well as the appropriate metadata sufficient to replicate their analysis. After depositing these data, authors should provide the appropriate accession numbers to look up the study and download data. These numbers should be referenced in the methods section of the text AND in the Plos One box asking authors to describe where data may be found.

Once these steps are completed or similar steps are taken that end in unrestricted access to the appropriate data, I will be happy to recommend this manuscript for immediate publication.

Reviewer #2: The authors addressed the reviewer comments well and I have no additional comments. Recommend publication.

Reviewer #3: Authors have responded to many of the previous concerns with few exceptions as noted below. In addition, there are some additional concerns related to changes made.

1. The objective remains unclear. Are the authors referring to articular cartilage (hyaline) in the first sentence or to elastic cartilage like in the ear. My previous concern was not fully resolved.

2. If authors prefer to use MRL instead of MRL/MpJ, this should be indicated at the first appearance as was suggested in the previous revision.

3. Authors have now included numeric data in the abstract. Author state that B6-vehicle mice healed ear hole punches poorly (0.25 0.03mm, ear hole healing 4 weeks after a 2mm ear hole punch, mean SEM), whereas MRL-vehicle mice healed well (1.4 0.1mm). Here and throughout the rest of the manuscript, authors showed that a lower value means poor healing and a higher value means better healing which is incorrect. A lower value is when the whole diameter is small (so better healing) and vice versa. Unless authors have used residual values by subtracting the hole diameter from the original diameter of 2 mm. This is not mention or is not clear. This information is important to correctly interpret the results.

4. Another related issue is the incorrect use of the term “hole closure rates” as authors did not actually measure the “rate,” they just measured the diameter of remaining hole. Please correct this throughout.

5. Values for adult mice (0.25 +/- 0.03) and for offspring are same. Please double check if this was the case or a typo?

6. What is exactly meant by “female microbiota resembled MRL mice” Do the authors mean “the microbiota from female mice resembled to that of MRL mice? If so, then indicate the sex of MRL mice.

7. Second sentence in Conclusion is not needed. Please consider deleting it.

8. Line 100-101: Please delete “conducted by the Sandell laboratory at Washington University” as the references have been included.

9. The response to my previous comment “It is not clear in methods how ear hole data was reported. A larger hole would mean less healing but that does not appear to be the case. This needs to be clarified to avoid confusion.” was not addressed.

10. Do authors have better images of ear holes. Why this figure was provided as supplemental.

11. Response to a previous comment about consistency of using same style is not fully addressed, for example ear hole vs earhole. Please take this opportunity to fix these minor issues.

12. Please proofread the final version as there are several extra spaces here and there.

7. PLOS authors have the option to publish the peer review history of their article (what does this mean?). If published, this will include your full peer review and any attached files.

Reviewer #1: No

Reviewer #2: No

Reviewer #3: No

---

## [Author Response · Author response to Decision Letter 1]

22 Jun 2021

Dear PLOS ONE editors and reviewers:

Thank you very much for your insightful and thorough review of our manuscript, entitled “Ear wound healing in MRL/MpJ mice is associated with gut microbiome composition and is transferable to non-healer mice via microbiome transplantation”. We have addressed each of the issues raised by the 3 reviewers in our revised manuscript and have detailed discussions of each below. We believe that these revisions have significantly improved the quality of our manuscript and look forward to your evaluation.

Editor Dr. David Burmeister, PhD:

You will need to upload your raw sequencing data, which is often done with NCBI's SRA function- https://www.ncbi.nlm.nih.gov/sra, and a statement in the manuscript including the link/accession number will be needed. This is recommended for OA journals, but specifically required given your NIH funding.

We apologize for this omission; the data have been submitted through SRA with the submission number PRJNA738540. Our manuscript has been updated with this information.

Additionally, reviewer 3 had some important clarifications- please indicate how you came up with the wound healing numbers (right now it is assumed that remaining wound was subtracted form the original 2mm size). As reviewer 3 points out, technically this is not rate since the data are not longitudinal, which would be interesting data to report. At minimum, clarity in the methods would help.

Agree, clarified throughout the article how wound healing numbers were calculated (2mm original size – final size). We actually went back and forth regarding the best way to present these data (as ‘healing’, larger = better healing, or as ‘final size’, smaller = better healing); in the end, we settled on reporting ‘healing’, as it makes the clade correlations we present later in the article easier to understand (a positive correlation equates to better healing and vice versa).

We also agree re: ‘rate’ and removed references to a healing rate as opposed to a final healing size.

Reviewer #1:

I am satisfied with changes made by the authors, however there is one item that needs to be addressed. In my previous review, I had mistakenly checked that data were available. The other two reviewers caught this correctly at that time. I do not see that data has been provided since then. For the Data Availability statement, the authors currently state that all data are available without restriction. The authors then point to the supplementary files which only contain some intermediate or raw result files of specific analyses. The standard in the field would be to publish the raw sequencing files and metadata on a database such as the Sequence Read Archive (SRA). At a minimum standard, authors need to provide raw sequencing files in fasta or fastq format as well as the appropriate metadata sufficient to replicate their analysis. After depositing these data, authors should provide the appropriate accession numbers to look up the study and download data. These numbers should be referenced in the methods section of the text AND in the Plos One box asking authors to describe where data may be found.

We apologize for this omission; the data have been submitted through SRA with the submission number SUB9866241. Our manuscript has been updated with this information.

Once these steps are completed or similar steps are taken that end in unrestricted access to the appropriate data, I will be happy to recommend this manuscript for immediate publication.

Reviewer #2: 

The authors addressed the reviewer comments well and I have no additional comments. Recommend publication.

Reviewer #3: 

Authors have responded to many of the previous concerns with few exceptions as noted below. In addition, there are some additional concerns related to changes made.

1. The objective remains unclear. Are the authors referring to articular cartilage (hyaline) in the first sentence or to elastic cartilage like in the ear. My previous concern was not fully resolved.

This article refers specifically to elastic cartilage. We have updated the manuscript to reflect this.

2. If authors prefer to use MRL instead of MRL/MpJ, this should be indicated at the first appearance as was suggested in the previous revision.

 Updated

3. Authors have now included numeric data in the abstract. Author state that B6-vehicle mice healed ear hole punches poorly (0.25 0.03mm, ear hole healing 4 weeks after a 2mm ear hole punch, mean SEM), whereas MRL-vehicle mice healed well (1.4 0.1mm). Here and throughout the rest of the manuscript, authors showed that a lower value means poor healing and a higher value means better healing which is incorrect. A lower value is when the whole diameter is small (so better healing) and vice versa. Unless authors have used residual values by subtracting the hole diameter from the original diameter of 2 mm. This is not mention or is not clear. This information is important to correctly interpret the results.

Our values of ‘healing’ represent (2.0mm – XX mm final earhole size). Updated the abstract, methods, and results section.

4. Another related issue is the incorrect use of the term “hole closure rates” as authors did not actually measure the “rate,” they just measured the diameter of remaining hole. Please correct this throughout.

 Removed ‘rate’ / ‘rates’

5. Values for adult mice (0.25 +/- 0.03) and for offspring are same. Please double check if this was the case or a typo?

 The values for offspring are actually listed correctly, added additional labels to further clarify.

6. What is exactly meant by “female microbiota resembled MRL mice” Do the authors mean “the microbiota from female mice resembled to that of MRL mice? If so, then indicate the sex of MRL mice.

By our statement ‘female microbiota resembled MRL mice’, we mean to say that the list of characteristic microbiome clades identified by LEfSe in female vs. male comparisons contain many of the same clades identified by LEfSe in MRL vs. B6 comparisons. 

We expanded our description of this comparison within the abstract to further clarify.

7. Second sentence in Conclusion is not needed. Please consider deleting it.

Deleted

8. Line 100-101: Please delete “conducted by the Sandell laboratory at Washington University” as the references have been included.

Deleted

9. The response to my previous comment “It is not clear in methods how ear hole data was reported. A larger hole would mean less healing but that does not appear to be the case. This needs to be clarified to avoid confusion.” was not addressed.

This is covered in response to issue #3 above.

10. Do authors have better images of ear holes. Why this figure was provided as supplemental.

We have cropped out extraneous portions of the image. This figure was provided as supplemental because we did not feel it provided readers any significant additional benefit over the data that were included in the manuscript; however, we have now included this within the body of the manuscript as Figure 2 and renumbered the other figures accordingly.

11. Response to a previous comment about consistency of using same style is not fully addressed, for example ear hole vs earhole. Please take this opportunity to fix these minor issues.

All changed to ‘ear hole’

12. Please proofread the final version as there are several extra spaces here and there.

Completed

Sincerely,

Matlock A. Jeffries MD FACP FACR

---

## [Editor Report · Decision Letter 2]

29 Jun 2021

Ear wound healing in MRL/MpJ mice is associated with gut microbiome composition and is transferable to non-healer mice via microbiome transplantation

PONE-D-21-05513R2

Dear Dr. Jeffries,

We’re pleased to inform you that your manuscript has been judged scientifically suitable for publication and will be formally accepted for publication once it meets all outstanding technical requirements.

Thank you for submitting your work to PLoS One, and congratulations on an interesting study.Kind regards,

David M. Burmeister, PhD

Academic Editor

PLOS ONE
---

## [Editor Report · Acceptance letter]

8 Jul 2021

PONE-D-21-05513R2 

Ear wound healing in MRL/MpJ mice is associated with gut microbiome composition and is transferable to non-healer mice via microbiome transplantation 

Dear Dr. Jeffries:

I'm pleased to inform you that your manuscript has been deemed suitable for publication in PLOS ONE. Congratulations! Your manuscript is now with our production department. 

Kind regards, 

on behalf of

Dr. David M. Burmeister 

Academic Editor

PLOS ONE